# Composite Multiscale Transition Permutation Entropy-Based Fault Diagnosis of Bearings

**DOI:** 10.3390/s22207809

**Published:** 2022-10-14

**Authors:** Jing Guo, Biao Ma, Tiangang Zou, Lin Gui, Yongbo Li

**Affiliations:** 1School of Mechanical Engineering, Beijing Institute of Technology, Beijing 100081, China; 2China North Vehicle Research Institute, Beijing 100072, China; 3MIIT Key Laboratory of Dynamics and Control of Complex System, School of Aeronautics, Northwestern Polytechnical University, Xi’an 710072, China

**Keywords:** composite multiscale transition permutation entropy, bearing, fault diagnosis, feature extraction

## Abstract

When considering the transition probability matrix of ordinal patterns, transition permutation entropy (TPE) can effectively extract fault features by quantifying the irregularity and complexity of signals. However, TPE can only characterize the complexity of the vibration signals at a single scale. Therefore, a multiscale transition permutation entropy (MTPE) technique has been proposed. However, the original multiscale method still has some inherent defects in the coarse-grained process, such as considerably shortening the length of time series at large scale, which leads to a low entropy evaluation accuracy. In order to solve these problems, a composite multiscale transition permutation entropy (CMTPE) method was proposed in order to improve the incomplete coarse-grained analysis of MTPE by avoiding the loss of some key information in the original fault signals, and to improve the performance of feature extraction, robustness to noise, and accuracy of entropy estimation. A fault diagnosis strategy based on CMTPE and an extreme learning machine (ELM) was proposed. Both simulation and experimental signals verified the advantages of the proposed CMTPE method. The results show that, compared with other comparison strategies, this strategy has better robustness, and can carry out feature recognition and bearing fault diagnosis more accurately and with improved stability.

## 1. Introduction

Rotating machinery is essential mechanical equipment which has been widely used in large-scale industries, such as aerospace, vehicle engineering, electrical engineering, machinery manufacturing, and so on. Bearings are an important part of electric and power transmissions, and a bearing fault is one of the main causes of rotating machinery faults [1,2]. Bearing fault diagnosis is vital for the healthy maintenance and reliable operation of rotating machinery. Bearing health detection can reduce the occurrence of rotating machinery failures, thus ensuring system safety and reducing maintenance costs [3].

In bearing health monitoring, vibration signal analysis is a commonly used fault feature extraction method. This is because the vibration signal contains a wealth of useful fault information [4]. In recent decades, bearing vibration signal processing and pattern recognition have become a research hotspot in the field of fault diagnosis. Kankar et al. [5] used an artificial neural network (ANN) and a support vector machine (SVM) to diagnose bearing faults, and verified that machine learning can be used for the automatic diagnosis of a bearing fault. With the development of deep learning algorithms, neural networks such as convolutional neural networks (CNN) have been effectively used in bearing fault diagnosis [6,7]. The authors of [8] proposed a deep learning model to preprocess the original signal for noise removal to overcome the shortcoming of the traditional intelligent method being greatly affected by noise. However, in practice, the vibration signal of the bearing has obvious nonlinear and non-stationary characteristics. Therefore, the analysis of nonlinear dynamic behavior and the extraction of useful and reliable fault features have become the key steps in fault diagnosis.

Entropy methods can quantify the dynamic trend and randomness of a nonlinear time series. In recent years, the use of entropy-based methods has become an important tool for analyzing signal complexity and feature extraction, and has been effectively used in fault diagnosis [9]. At present, approximate entropy (AE), sample entropy (SE), permutation entropy (PE), fuzzy entropy (FE), and diversity entropy (DE) methods are widely used in fault diagnosis of rotating machinery. AE was proposed by Pincus [10], and can be used to measure the regularity of a time series. Richman et al. [11] proposed SE, which uses the association dimension to induce SE to show relative consistency, and its complexity analysis performance is better than that of AE. However, SE has the disadvantage of relying heavily on data length [12]. The FE method proposed by Chen et al. [13] is an improvement on the SE method. Other researchers [14] have used FE to measure the complexity of vibration signals, and they verified the excellent dynamic tracking performance of FE and its ability to obtain a more accurate complexity estimation. In consideration of noise resistance and computational efficiency, Wang et al. [15] proposed DE, which uses cosine similarity to measure the divergence of orbits. Bandt et al. [16] proposed PE to calculate the state probability of track arrangement order, showing it has high computational efficiency and good feature extraction effects in signal processing.

In order to overcome the problem of insufficient information analysis when using SE to evaluate the dynamic characteristics and randomness of complex data, Costa et al. [17] proposed using multiscale sample entropy (MSE) to evaluate the complexity of time series over a range of scales. MSE has been successfully applied to analyze vibration signals generated by various dynamic behaviors [18,19,20]. Based on the same coarsening process as MSE, FE, PE, and DE can be extended to multiscale fuzzy entropy (MFE) [21,22,23], multiscale permutation entropy (MPE) [24,25,26], and multiscale diversity entropy (MDE) [15]. Through coarse-grained processing, the original time series can be divided into several short time series. The coarse-grained time series can represent the dynamic distribution characteristics of the original signal at a certain scale. Therefore, multiscale processing enhances the performance of entropy methods in evaluating signal complexity. On a multiscale basis, the combination of the symbol dynamic filtering process and the entropy method can not only remove noise, but also significantly improve the computational efficiency and feature extraction ability [27,28,29].

Recently, Zhang et al. [30] proposed a novel complexity estimation method, transition permutation entropy (TPE). TPE is different from the other methods in that it extracts the features of a time series from the transition probability matrix of ordinal patterns. Because the eigenvalue is very important when analyzing the dynamic behavior, TPE uses the positive eigenvalue of the transition probability matrix to calculate the entropy. This improves the feature identification performance of a time series. However, TPE only analyzes a time series using a single scale, which reduces the accuracy and comprehensiveness of the information analysis. Therefore, in this work we extended TPE to multiscale analysis. In the traditional multiscale calculation method, the coarse-grained time series is obtained by calculating the arithmetic mean of adjacent data points on the original time series without overlapping. The length of the coarse-grained time series obtained in this way is too short at large scale, and the accuracy and stability will be affected. Therefore, in this work, composite multiscale transition permutation entropy (CMTPE) was proposed as a way to solve these obstacles. When the composite multiscale method is used to coarse-grain the original time series, a coarse-grained time series with different starting points can be obtained at each scale, and the number is equal to the scale factor. Each coarse-grained time series can characterize the dynamic characteristics and randomness of the original signal, which can effectively enhance the accuracy and stability of TPE. CMTPE not only had excellent feature extraction performance, but also had better robustness to noise. The superiority of the proposed CMTPE method was verified by the simulation and experimental signals of bearing faults. The main contributions of this study are given as follows:(1)In order to enhance the accuracy of feature extraction and the comprehensiveness of information analysis, CMTPE was proposed as a strategy to quantify the complexity of time series.(2)A fault diagnosis strategy based on CMTPE and ELM was proposed for bearing fault type identification to identify the fault types of bearings.(3)The advantages of CMTPE in feature extraction were verified by simulation and experimental signals. Comparing TPE, MPE, and MTPE, the results showed that CMTPE has the highest fault diagnosis performance.

The rest of this paper is organized as follows: The concept of CMTPE is introduced in Section 2. In Section 3, results from simulation signals used to validate the superiority of CMTPE are reported. In Section 4, the effectiveness of CMTPE was verified using experimental signals. Finally, the conclusion of this article is provided in Section 5.

## 2. Methodology

In this section, the theories of TPE and MTPE are introduced in detail. In addition, the concept of the CMTPE algorithm is proposed.

### 2.1. Transition Permutation Entropy (TPE)

A time series of length N can be written as $X=\left\{x_{1},x_{2},\cdots ,x_{N}\right\}$. The TPE algorithm is introduced as follows:

Step 1. According to the phase space embedding theory, reconstruct X into a series of vectors with embedding dimension m. The reconstructed phase space is as follows:(1)$$X=\left[\begin{array}{cccc} x_{1} & x_{2} & \cdots & x_{N-m+1}\\ x_{2} & x_{3} & \cdots & x_{N-m+2}\\ \vdots & \vdots & \ddots & \vdots \\ x_{m} & x_{m+1} & \cdots & x_{N} \end{array}\right]$$

The reconstructed vectors can be expressed as $X_{i}=\left\{x_{i},x_{i+1},\cdots ,x_{i+m-1}\right\}$, 1≤i≤N−m+1.

Step 2. Compare the size relationship of the elements in the vector, so as to identify the ordinal pattern of each vector. When the embedding dimension is m, there are m! possible ordinal patterns for any vector. For example, if the embedding dimension m=3, there are 6 ordinal patterns. The size relationship of all vectors can be expressed by the size relationship of 0, 1, 2. For vector $\left(x_{k-1},x_{k},x_{k+1}\right)=\left(18,3,15\right)$, the element size relationship is $x_{k}<x_{k+1}<x_{k-1}$, and its corresponding ordinal pattern is π=2,0,1.

Step 3. Calculate the transition probability between the ordinal patterns corresponding to all vectors to obtain the following transition probability matrix P:(2)$$P=\left[\begin{array}{cccc} p_{11} & p_{12} & \cdots & p_{1n}\\ p_{21} & p_{22} & \cdots & p_{2n}\\ \vdots & \vdots & \ddots & \vdots \\ p_{n1} & p_{n2} & \cdots & p_{nn} \end{array}\right]$$
where n=m!, and $p_{ij}$ represents the probability of an transition from pattern $\pi_{i}$ to pattern $\pi_{j}$.

Step 4. Calculate the TPE using the positive eigenvalue of matrix P. If the eigenvalue is complex number, its real part is taken. If P has n positive eigenvalues $\lambda_{i}$, TPE is calculated as follows:(3)$$\mathrm{TPE}\left(X,m\right)=-\sum_{i=1}^{n}\frac{\lambda_{i}}{m!}\log\frac{\lambda_{i}}{m!}$$

### 2.2. Multiscale Transition Permutation Entropy (MTPE)

The entropy calculated from a single scale can only provide poor fault information. Multiscale analysis can extract more useful information from time series of different scales. The MTPE algorithm consists of two steps: (1) Conducting a coarse-graining process to obtain the series of the original time series at different scales; and (2) Calculating the TPE of each coarse-grained time series. First, divide the time series $X=\left\{x_{1},x_{2},\cdots ,x_{N}\right\}$ into multiscale time series $Y=\left\{Y_{1},Y_{2},\cdots ,Y_{\tau }\right\}$. The scale factor τ is a positive integer. The time series at any scale is $Y_{\tau }=\left[y_{1,\tau },y_{2,\tau },\cdots ,y_{j,\tau }\right]$, j=N/τ, and the calculation is as follows:(4)$$y_{s,\tau }=\frac{1}{\tau }\sum_{i=\tau \left(s-1\right)+1}^{\tau s}x_{i}$$

Then, the time series of all scales obtained from the above process can be substituted into the TPE algorithm to calculate the MTPE as follows:(5)$$\mathrm{MTPE}\left(X,m,\tau \right)=\mathrm{TPE}\left(Y_{\tau },m\right)$$

### 2.3. Composite Multiscale Transition Permutation Entropy (CMTPE)

In order to further improve the accuracy and stability of MTPE, CMTPE was proposed. When the scale factor is τ, τ different time series can be obtained. MTPE only considers the first coarse-grained time series at each scale, while CMTPE considers all τ coarse-grained time series. As shown in Figure 1, when the scale factor τ=3, MTPE only calculates one coarse-grained time series $y_{1}^{\left(3\right)}$, while CMTPE calculates three coarse-grained time series $y_{1}^{\left(3\right)}$, $y_{2}^{\left(3\right)}$, and $y_{3}^{\left(3\right)}$. Divide the time series $X=\left\{x_{1},x_{2},\cdots ,x_{N}\right\}$ into multiscale time series $Y=\left\{Y_{1},Y_{2},\cdots ,Y_{\tau }\right\}$. The time series at any scale is $Y_{\tau }=\left[y_{\tau }^{1},y_{\tau }^{2},\cdots ,y_{\tau }^{k},\cdots ,y_{\tau }^{\tau }\right]$, where $y_{\tau }^{k}=\left[y_{1,\tau }^{k},y_{2,\tau }^{k},\cdots ,y_{j,\tau }^{k}\right]$. The calculation is as follows:(6)$$y_{j,\tau }^{k}=\frac{1}{\tau }\sum_{i=\tau \left(j-1\right)+k}^{\tau j+k-1}x_{i}$$

CMTPE is the mean of the TPE values for all coarse-grained time series, that is,
(7)$$\mathrm{CMTPE}\left(X,m,\tau \right)=\frac{1}{\tau }\sum_{k=1}^{\tau }\mathrm{TPE}\left(y_{\tau }^{k},m\right)$$

CMTPE considers all τ different coarse-grained time series, at each scale factor τ. Therefore, CMTPE can extract more fault information from the original time series. The entropy calculated by this method is more accurate and stable than that calculated by MTPE.

.

### 2.4. CMTPE Based Fault Diagnosis Strategy

In this work, a fault diagnosis strategy based on CMTPE was proposed. In this strategy, an ELM classifier was used to identify different fault types. The overall fault diagnosis framework is shown in Figure 2 [31]. The main steps were as follows:

Step 1. The vibration signals of bearings under different health conditions are measured by sensors.

Step 2. CMTPE is used for feature extraction of vibration signals. Each health condition will provide the corresponding entropy characteristics, representing the complexity of different vibration signals.

Step 3. A part of the fault features is randomly selected as a training set to train the ELM classifier.

Step 4. The remaining features are used as the test set to test the trained ELM, and the fault recognition rate is obtained. Steps 3 and 4 are run 20 times to obtain the average test accuracy.

## 3. Simulation Evaluation

### 3.1. Simulated Bearing Signal

In this section, in order to verify the effectiveness and advantages of the proposed CMTPE, we detail the three types of simulated bearing faults which were designed: outer race fault, inner race fault, and ball fault models. The schematic diagram of the three simulated faults is shown in Figure 3.

In the load area, as shown in Figure 3, the sensor was installed at the maximum load density. Figure 3a shows the fault model of an outer race fault. Since the location of a localized defect will not change with time, the impulse force can be regarded as an ideal force. Figure 3b shows the fault model of an inner race fault, which has the same basic assumptions as the outer race fault model. At the peak of the load area, the ball will contact with a localized defect, resulting in the first impulse. After that, the localized defect will rotate with the inner race, so the contact position between the ball and the inner race will change with time. This type of contact will generate an impulse force only when it occurs in the load area. Figure 3c shows the ball fault model, which also has the same basic assumptions as the outer race fault model. In contrast to the inner race fault, a localized defect will rotate with the ball, and the defect will continuously contact the inner and outer races to continuously generate impulse force [32].

The simulated bearing type was an N205 cylindrical roller bearing. The rotating speed was 3000 rpm. The sampling frequency was 10,240 Hz. The detailed bearing dimensions are shown in Table 1.

The fault frequency can be calculated according to the parameters in Table 1. Main parameters of the bearing: roller diameter d=6.5mm; pitch circle diameter D=35.5mm; number of rollers Z=12; contact angle α=0; rotating speed v=3000rpm. The fault frequency was calculated as follows:

(1) Outer race fault characteristic frequency $f_{0}$
(8)$$f_{0}=\frac{1}{2}Z\left(1-\frac{d}{D}\cos\alpha \right)\frac{v}{60}=245.0704(\mathrm{Hz})$$

(2) Inner race fault characteristic frequency $f_{i}$
(9)$$f_{i}=\frac{1}{2}Z\left(1+\frac{d}{D}\cos\alpha \right)\frac{v}{60}=354.9296\left(\mathrm{Hz}\right)$$

(3) Ball fault characteristic frequency $f_{e}$
(10)$$f_{e}=\frac{D}{d}\left(1-{\left(\frac{d}{D}\right)}^{2}\cos^{2}\alpha \right)\frac{v}{60}=263.9220\left(\mathrm{Hz}\right)$$

Figure 4 shows the time domain and envelope spectrum of each of the three simulated fault types. Among the data, Figure 4a,c,e depicts the time domain diagrams of the three faults, and Figure 4b,d,f shows the corresponding envelope spectrum diagrams. The fault frequency is marked with a blue arrow in the envelope spectrum.

### 3.2. Analysis of Simulation Results

In the practical working environment, the operation of the equipment is influenced by noise. Therefore, the simulated bearing fault signal was added to Gaussian white noise with different signal-to-noise ratios (SNR) to simulate the actual working conditions. SNR ranged from 10 dB to 40 dB, in 1 dB steps.

In this simulation, MTPE, TPE, MPE, and the proposed CMTPE were used to extract fault features from simulation signals. For the selection of the main parameters when using the above method, there were the following considerations: if the embedding dimension m is small, the dynamic process of reconstruction will contain non detailed dynamic information, while if the value of m is too large, the number of vectors will decrease, which will lead to the loss of information. In addition, a large value of the scale factor τ will lead to information redundancy, and a small value of τ will lead to the loss of fault information. Therefore, the recommended value for parameter τ is 10–20 [29]. The values for the parameters of the entropy methods used in this study were set as m=3 and τ=20.

The fault diagnosis strategies of each of the four methods combined with ELM were used to identify three simulated bearing faults. For each fault type, the original signal was sliced into 100 samples without overlap, and the data length of each sample was 2048. Therefore, the data set had a total of 300 samples. Among them, 50 samples of each fault type were randomly selected as the ELM training set, and the rest of the samples were used to test the trained ELM. The ELM was run 20 times and the average test accuracy was taken as the final test accuracy. Higher test accuracy means better performance of fault diagnosis strategy, and smaller test variance means better stability. The robustness of the method against noise can be obtained by comparing the test accuracies for each different SNR value.

The test results are shown in Figure 5. It is obvious that no matter the SNR value, the test accuracy of CMTPE was always higher than that of the other methods, and its error bar was also smaller than that of other methods. This shows that CMTPE had the best bearing fault diagnosis performance and the highest test stability. When the signal-to-noise ratio was high, all strategies except TPE had high accuracy. For example, when the SNR range was 30 to 40, the test accuracy of CMTPE, MTPE, and MPE was higher than 95%. However, as the SNR gradually decreased, the test accuracy of MTPE and MPE decreased at a faster speed and a larger range, while CMTPE still had 100% test accuracy, even when the SNR value decreased to 20 dB. Moreover, when the SNR was reduced to 10 dB, the test accuracy of MTPE and MPE was only 61.90% and 71.73%, respectively, while the test accuracy of CMTPE was still 89.60%. This shows that the proposed CMTPE method has better robustness to noise.

When the SNR was less than 5, the classification accuracy of all methods was less than 60%, and the accuracy near 0 was less than 50%. When the SNR was negative, the accuracy of the classifier was no longer referential, because at this time, the simulation signal had been submerged by noise, and none of the four methods could correctly identify faults. However, when the noise was relatively weak, CMTPE still had the highest classification accuracy and the smallest error bar, compared with other methods, and had better stability. In the case of negative SNR, we used filtering and other noise reduction methods to preprocess the signal to achieve better anti-noise effects.

## 4. Experimental Evaluation

In this section, we report the testing of the effectiveness of the CMTPE using bearing fault data. CMTPE was compared with TPE, MTPE, and MPE to verify the superiority of the CMTPE-based fault diagnosis strategy.

### 4.1. Bearing Test Rig and Experimental Data Illustration

The experimental data were collected on an HD-FD-H-03X rotor rolling bearing fault test rig. The appearance and structure of the platform are shown in Figure 6. In the experiment, the speed of the motor was 1000 rpm and no load was applied. In order to verify the effectiveness of the proposed method, five different health conditions were designed, as shown in Figure 7. One of them was designated as normal, and the other four fault types were inner race crack 4mm (IRC), outer race crack 4 mm (ORC), inner race pitting 3 mm (IRP), and outer race pitting 3 mm (ORP).

The vibration signals of the different health conditions were collected through the acceleration sensor, in which the sampling frequency was 10,240 Hz. Figure 8 shows the time domain and envelope spectrum of each of the four fault states. Figure 8a,c,e,g shows the time domain diagrams of three faults; Figure 8b,d,f,h displays the corresponding envelope spectrum diagrams. The vibration signal of each state was divided into 75 samples for feature extraction, and the length of each sample was 2048. Then, 25 samples of each state were randomly selected to train the ELM classifier, and the remaining samples were used for testing [33]. Therefore, the total number of training and test sets was 125 and 250, respectively.

### 4.2. Comparison Analysis

According to the proposed fault diagnosis strategy, CMTPE was used to extract the fault features of bearing vibration signals. In addition, in order to prove that CMTPE has better feature extraction and fault diagnosis ability, MPE, TPE, and MTPE were used for comparison. The main parameters of these methods were as follows: the scale factor of the multi-scale methods was τ=20 and the embedding dimension of all methods was m=3. The features extracted by the above four methods were used to train and test the ELM. In order to reduce the error caused by randomness, each method was run 20 times, and then the average test accuracy was taken as the final classification result. The test accuracy of the four strategies over 20 runs is shown in Figure 9. Table 2 shows the classification accuracy and variance. The criteria were: higher accuracy represents better feature extraction ability, and lower variance represents better stability.

From Table 2 and Figure 9, it can be observed that the fault identification accuracy of TPE was lower than that of the other multiscale methods, and that the identification accuracy of CMTPE reached 98.60%, which was the highest among the multiscale methods. This shows that the multiscale analysis could extract more abundant fault information when processing vibration signals. Moreover, the coarse-grained process also affects the quality of the fault features. The higher identification accuracy proves that the coarse-grained process of the proposed CMTPE method can better grasp the key information related to bearing faults, and it had the best feature extraction effect. Furthermore, the variance of CMTPE was only 0.65%, which was lower than the other methods. This verifies that the proposed fault diagnosis strategy based on CMTPE not only had excellent fault feature extraction performance, but also had the best stability.

As shown in Figure 10, the confusion matrix of the four methods can intuitively visualize the classification performance of each method combined with ELM [34,35]. As shown in Figure 10b,c, TPE and MTPE exhibited many misclassifications; in particular, it was difficult to distinguish between IRC and IRP, while, as shown in Figure 10a,d, CMTPE and MPE have better classification performance. CMTPE had only four misclassifications, and the classification accuracy reached 98.4%, which was the highest among all methods.

In this work, 25 training samples and 50 test samples were selected. In order to eliminate the contingency brought by the specific number of training samples, a performance test of CMTPE with different numbers of training samples was carried out. The classification accuracy was tested with 15, 25, 35, 45, 55 and 65 training samples, and each case was run 20 times to reduce randomness. The results are shown in Figure 11. Obviously, with an increase in the number of training samples, the accuracy of various methods will increase, but CMTPE always had the highest test accuracy compared to the other methods.

In order to further intuitively compare the feature extraction capabilities of the three multiscale methods, we carried out visual processing on the extracted fault features. In this work, the scale factor of multiscale method was τ=20. Therefore, the t-SNE visualization method was used to reduce the dimension of fault features to two dimensions [36]. The results of this feature visualization are shown in Figure 12. The criterion of feature extraction effect is: the closer the distance between clusters of the same type of features, the farther the distance between clusters of different types of features, which proves that the feature extraction effect of this method is better.

As can be seen from Figure 12b, the features of the states, other than ORC, were obviously mixed, which indicates that MTPE has poor feature extraction performance. Figure 12a demonstrates that the features extracted by MPE could better distinguish between most fault characteristics. However, the distance between clusters of different state features was small, some feature points were mixed, and the distance within clusters was large, so the effect of feature extraction were poor. In contrast, it can be seen from Figure 12c that CMTPE had the largest inter-cluster distance and the smallest intra-cluster distance, and that the feature extraction performance was the best.

In order to test the minor fault identification ability of the proposed method, the following two cases were designed with different extents of faults. Case 1 included a normal control and eight different degrees of inner and outer race crack faults: normal, inner race crack 0.2 mm, outer race crack 0.2 mm, inner race crack 1 mm, outer race crack 1 mm, inner race crack 2.7 mm, outer race crack 2.7 mm, inner race crack 4 mm, and outer race crack 4 mm. Case 2 included a normal control and six different degrees of pitting faults: normal, inner race pitting 1 mm, outer race pitting 1 mm, inner race pitting 2 mm, outer race pitting 2 mm, inner race pitting 3 mm, and outer race pitting 3 mm.

The vibration signals of each state were again divided into 75 samples, with 25 used for training the ELM, and the rest used for testing. The classification accuracy and variance under the two cases for the four methods are shown in Table 3 and Table 4. It was found that CMTPE still had an excellent classification effect for the more minor faults, especially for pitting faults of different degrees; the test accuracy reached 99.67%. CMTPE also still had the highest stability. However, the other three methods had decreased test accuracy for minor faults, in particular, the classification effect of TPE was very poor. This also shows that the composite multiscale method can avoid the information loss caused by the single scale method, and also overcome the problem of low accuracy of entropy estimation caused by the traditional multiscale method. Therefore, CMTPE can effectively identify minor faults.

## 5. Conclusions

In this study, a method using CMTPE for quantifying the complexity of time series was proposed. CMTPE takes into consideration the transition probability matrix of an ordinal pattern and performs composite multiscale processing on the original time series. This avoids the loss of information caused by single-scale analysis, and overcomes the problem where the traditional multiscale method will greatly shorten the time series in large scale, resulting in low accuracy of entropy evaluation. Composite multiscale analysis improved the performance of CMTPE feature extraction, the accuracy of entropy estimation, and the robustness against noise. Compared with MTPE, TPE, and MPE, the superiority of CMTPE was verified by both simulation and experimental data. The results show that CMTPE has better robustness, can effectively identify bearing faults, and has the highest diagnostic accuracy and stability.

Moreover, in the case of negative SNR, it was necessary to use filtering and other noise reduction methods to preprocess the signal to achieve better anti-noise effect. Thus, in future work, we will test the effectiveness of combining CMTPE with other noise reduction methods.

## Figures and Tables

**Figure 1 sensors-22-07809-f001:**
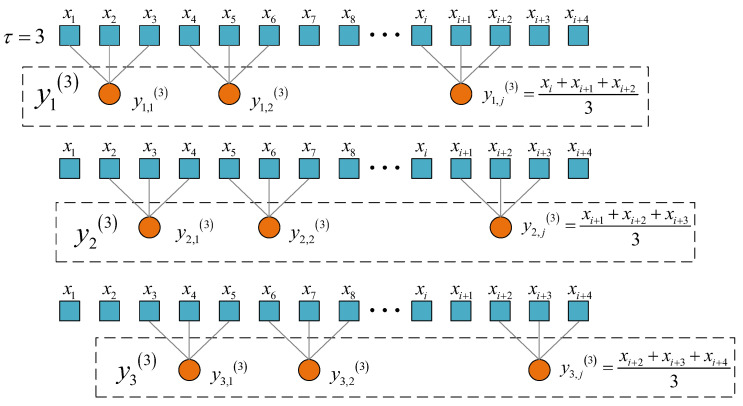
Schematic diagram of multiscale coarsening process when scale factor τ=3.

**Figure 2 sensors-22-07809-f002:**
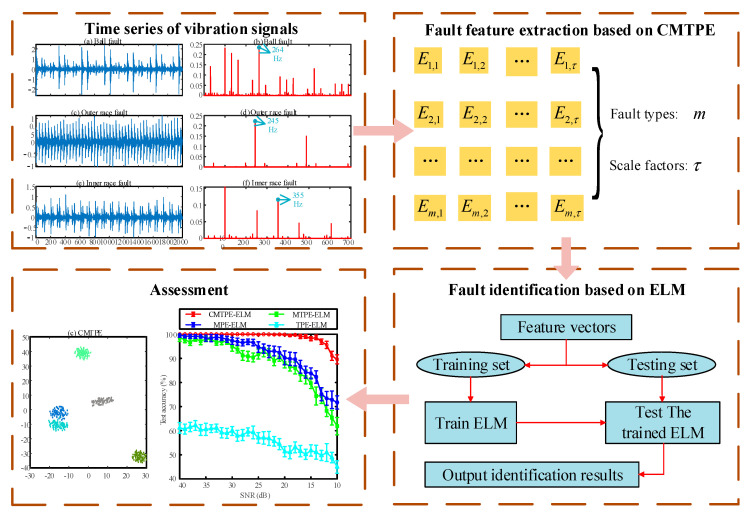
The overall fault diagnosis framework.

**Figure 3 sensors-22-07809-f003:**
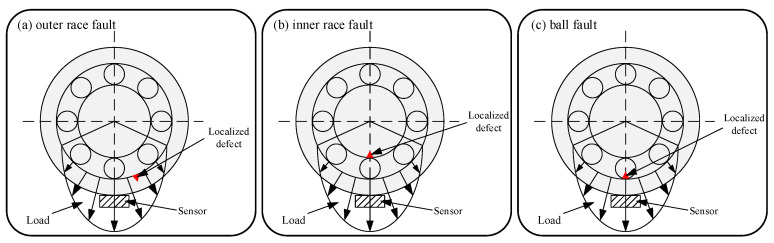
The schematic of simulated bearing faults.

**Figure 4 sensors-22-07809-f004:**
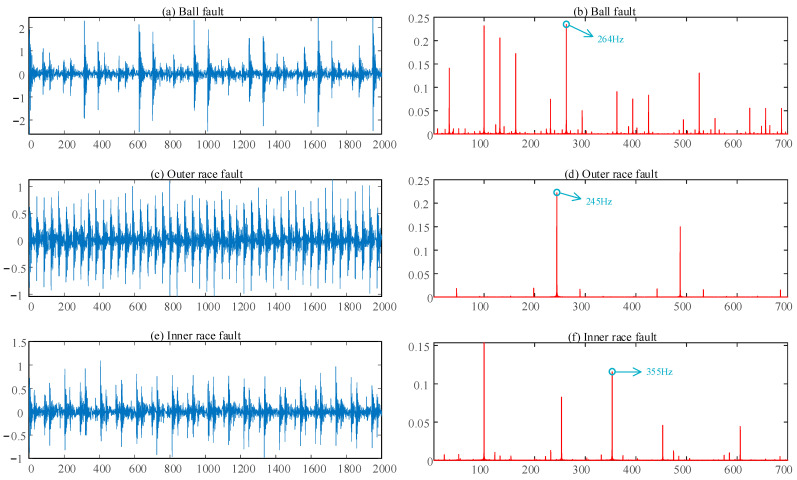
Time domains and spectra of simulated bearing faults: (**a**) time domain of ball fault; (**b**) envelope spectrum of ball fault; (**c**) time domain of outer race fault; (**d**) envelope spectrum of outer race fault; (**e**) time domain of inner race fault; (**f**) envelope spectrum of inner race fault.

**Figure 5 sensors-22-07809-f005:**
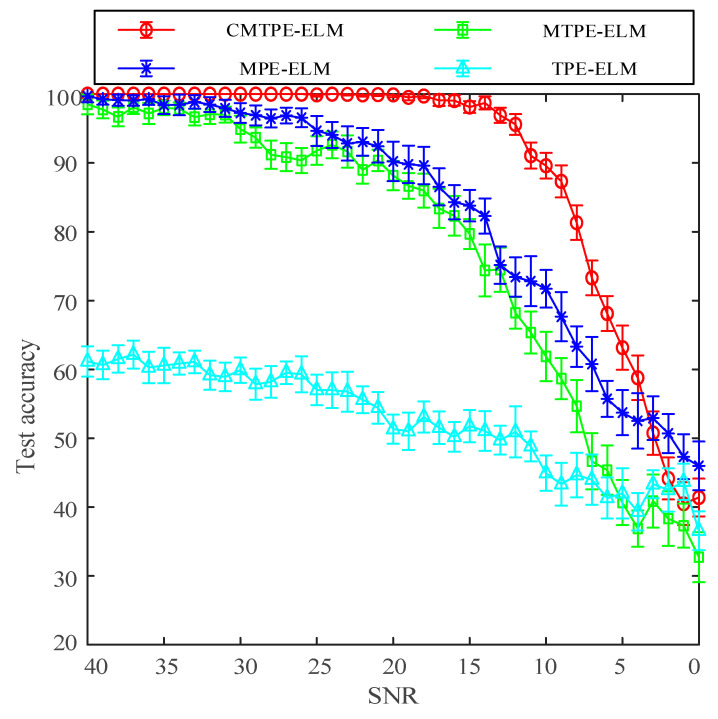
Test results with different SNR value.

**Figure 6 sensors-22-07809-f006:**
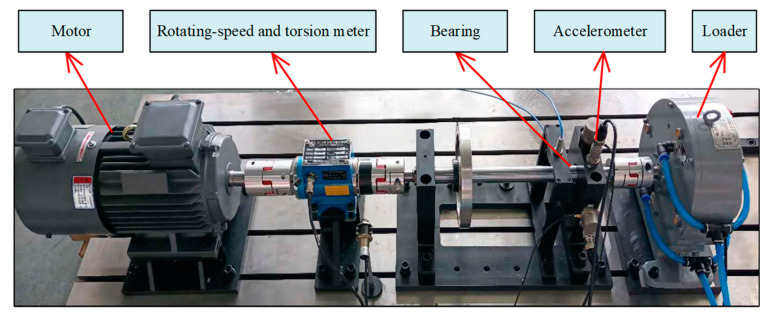
The bearing fault test rig used in the experiment.

**Figure 7 sensors-22-07809-f007:**
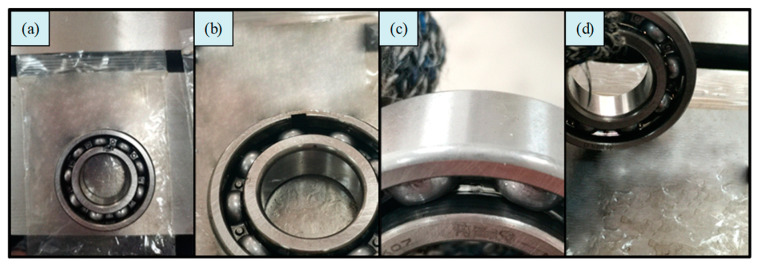
Four types of bearing fault designed in the experiment: (**a**) IRC; (**b**) ORC; (**c**) IRP; (**d**) ORP.

**Figure 8 sensors-22-07809-f008:**
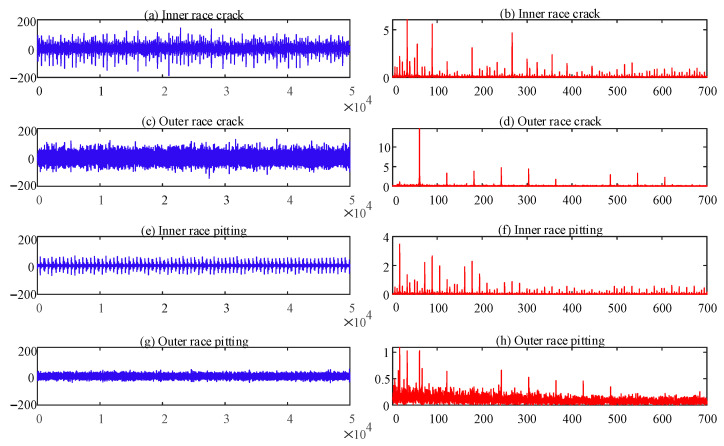
Time domains and spectra of the four fault states: (**a**) time domain of IRC; (**b**) envelope spectrum of IRC; (**c**) time domain of ORC; (**d**) envelope spectrum of ORC; (**e**) time domain of IRP; (**f**) envelope spectrum of IRP; (**g**) time domain of ORP; (**h**) envelope spectrum of ORP.

**Figure 9 sensors-22-07809-f009:**
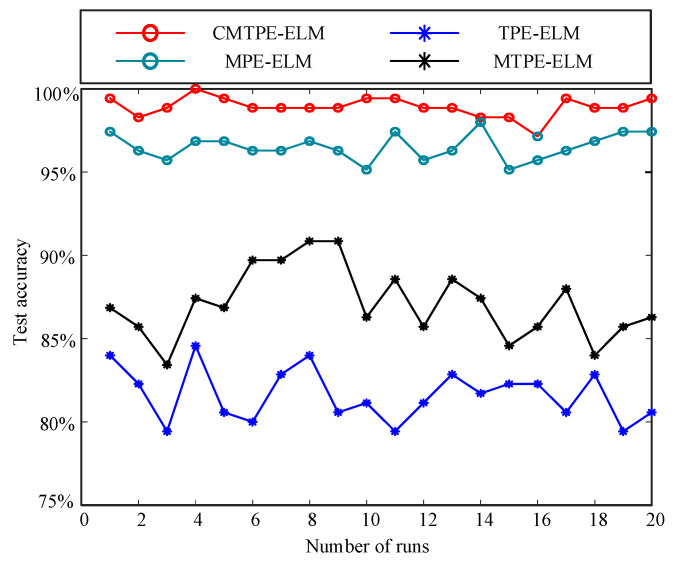
Test accuracy of the four strategies.

**Figure 10 sensors-22-07809-f010:**
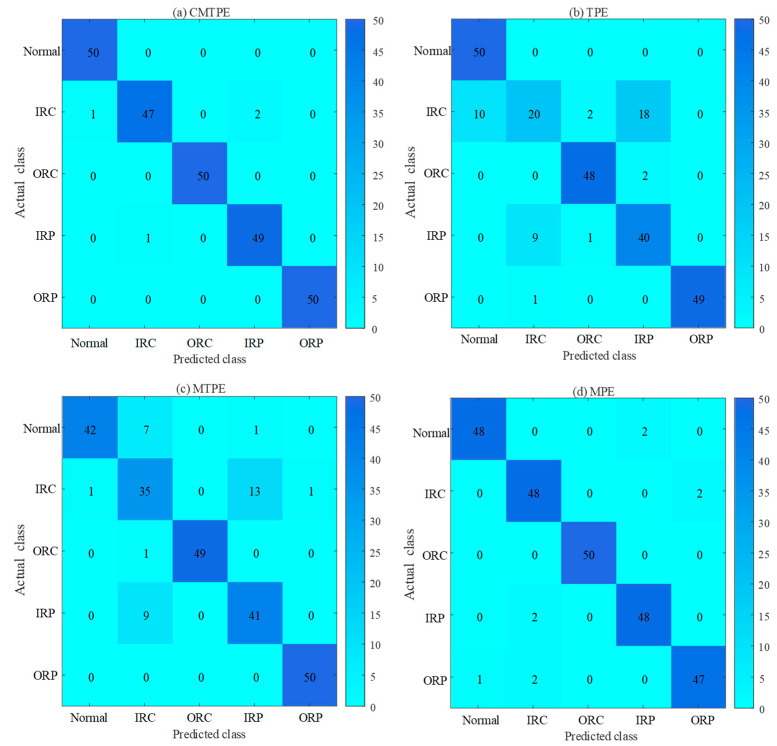
Confusion matrix of four methods.

**Figure 11 sensors-22-07809-f011:**
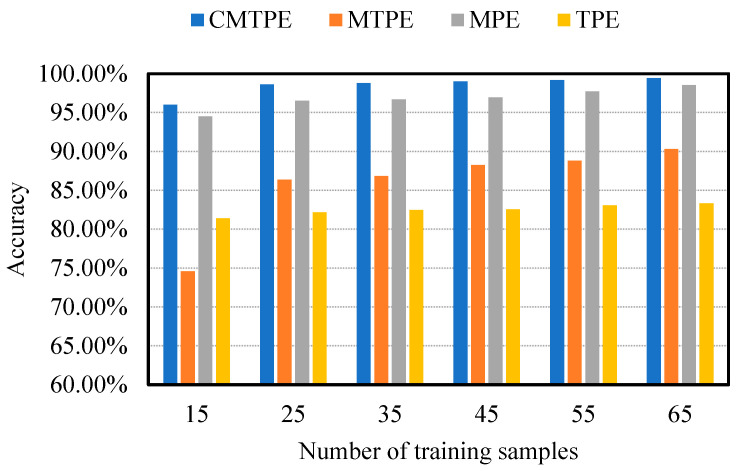
Effect of the number of training samples on the performance of CMTPE, MTPE, MPE and TPE.

**Figure 12 sensors-22-07809-f012:**
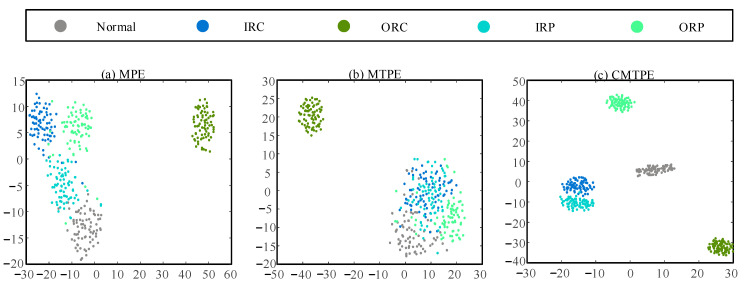
Visualization of features extracted by three multiscale entropy methods: (**a**) MPE method; (**b**) MTPE method; (**c**) CMTPE method.

**Table 1 sensors-22-07809-t001:** Bearing parameters.

Parameter	Value
Pitch circle diameter	35.5 mm
Roller diameter	6.5 mm
Rotating speed	3000 rpm
Number of rollers	12
Sample frequency	10,240 Hz
Natural frequency of bearing	4000 Hz
Contact angle	0°

**Table 2 sensors-22-07809-t002:** Average test accuracy and variance of the four methods.

Methods	CMTPE	TPE	MTPE	MPE
Average test accuracy (%)	98.60	82.16	86.37	96.51
Variance (%)	0.65	1.39	1.92	0.94

**Table 3 sensors-22-07809-t003:** Average test accuracy and variance of the four methods for different degrees of crack faults.

Methods	CMTPE	TPE	MTPE	MPE
Average test accuracy (%)	96.46	44.71	70.46	94.31
Variance (%)	0.74	1.95	1.85	1.01

**Table 4 sensors-22-07809-t004:** Average test accuracy and variance of the four methods for different degrees of pitting faults.

Methods	CMTPE	TPE	MTPE	MPE
Average test accuracy (%)	99.67	60.49	86.39	94.97
Variance (%)	0.34	1.51	1.90	1.06

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
