# Peer review of "Composite Multiscale Transition Permutation Entropy-Based Fault Diagnosis of Bearings"

_sensors, 2022, doi:10.3390/s22207809_

Round 1

Reviewer 1 Report

It is a good paper, I just want to ask a few questions before proceeding.

1. Does this approach applicable for bearings with clearance C2 and C3?

2.  Can these results be replicated across bearings of large scale industrial motors? Means the effectiveness of CMTPE in processing signal at a large scale? 

3. These are a few formatting issues that must be addressed accordingly. 

Reviewer 2 Report

This paper proposed an effective bearing diagnostic method that integrates a composite multiscale transition permutation entropy and extreme learning machine. Comments about the paper are as follows:

(1) The fault characteristic frequencies of the bearing should be mentioned in this manuscript and marked in Fig. 3. Which criterions are based on the selection of the embedding dimension and the scale factor?

(2) When the value of SNR is greater than zero, it indicates that the power of the signal is higher in comparison to the power of the noise (see Fig.4). The negative SNRs are required to obtain a better evaluation of the proposed noise-insensitive CMTPE algorithm.

(3) In section 4.2, confusion matrices should be provided to show the performances of ELM with MPE, MPE and CMTPE.

(4) Please recheck the sentence (see Line 56).

Reviewer 3 Report

·        In figure 7, keep the scale, maximum and minimum limit of the Y axis the same for all plots.

·        Classification results must be presented using a confusion matrix at least once. You refer to this article https://www.sciencedirect.com/science/article/pii/S0263224120311659

·        What is the possibility of misclassification of a faulty condition as healthy depending on the degree of fault? If the model is deployed in real-time and such a situation arises, how will you identify that the tool is in the failure zone and showcased as healthy by your TCM system?

·        What about the classification of tool fault using blind data (No labels)? How will you address this? Suggest a methodology by applying the trained model to classify blind datasets. Refer to "Figure 8: Framework for classification of blind data" from the following article. DOI is https://doi.org/10.36001/ijphm.2020.v11i2.2929

·        Add the pictorial flowchart of the overall framework which assists readers in understanding the overall framework. You may refer to the following reference. DOI is https://doi.org/10.32604/sv.2022.014910 

·        How was the ratio of training and testing considered? Additionally, results must be provided considering other holdout % and holdout validation approaches. Refer to this article to understand the holdout validation approach. DOI is https://doi.org/10.1115/1.4051696

Round 2

Reviewer 2 Report

(1)   The calculation of the fault characteristic frequencies of the bearing (see Fig.4) should be elaborately described in the revised manuscript.

(2)   The spectra of Fig.8 should be given.

    (3)   The Fig. 5 (see Response to Reviewers' Comments) can replace the Fig. 5            in the revised manuscript. And the authors should describe the                          limitation of the proposed method. Please explain it in the conclusion.

Reviewer 3 Report

Even though the authors have tried to answer my comments, I am least convinced with the response, and are not appropriate.

Also following major concerns should be addressed.

1. In Figure 10, the confusion matrix for case TPE, there is a major concern of Type II error for IRC and normal cases. Only 20 samples of IRC are correctly classified. However, 10 samples are classified as a normal condition. This is the case of Type II error where the faulty condition is classified as healthy which is very dangerous. This concern should be addressed thoroughly.

2. In addition to this, many faulty conditions are misclassified as other faulty conditions. See Figure 10: MTPE case, 9 samples of IRP are misclassified as IRC. How will you address this?

3. How will you deal with misclassification depending on the degree of fault? This question was asked in the first review also but the authors’ response is not appropriate.

4. The methods/review/contents included by the authors in the revision as suggested by me have not been cited. For example, regarding the methodology of classification of blind data, the following references should be cited.

Application of bayesian family classifiers for cutting tool inserts health monitoring on CNC milling, A Bayesian Optimized Discriminant Analysis Model for Condition Monitoring of Face Milling Cutter Using Vibration Datasets, A machine learning approach for vibration-based multipoint tool insert health prediction on vertical machining centre (VMC), Application of Machine Learning for Tool Condition Monitoring in Turning

A further major revision according to my comments is needed for final acceptance. There is a huge concern about the novelty of the work.
